# Urban Tree Canopy and Environmental Justice: Examining the Distributional Equity of Urban Tree Canopy in Guangzhou, China

**DOI:** 10.3390/ijerph20054050

**Published:** 2023-02-24

**Authors:** Yutian Zhuang, Dixiang Xie, Xijun Yu

**Affiliations:** 1School of Architecture and Urban Planning, Guangdong University of Technology, Guangzhou 510090, China; 2South China Institute for Environmental Science, Ministry of Ecology and Environment, Guangzhou 510655, China

**Keywords:** urban tree canopy, environmental justice, house price, urban green space, Guangzhou

## Abstract

Rapid urbanization has led to a series of environmental problems that are harmful to the physical and mental health of residents. Not only can increasing urban tree canopy (UTC) improve sustainable city development, but it can also effectively promote the quality of life for residents; however, the uneven spatial distribution of UTC can also bring about social justice problems. Currently, there are few studies related to the justice of UTC distribution in China. Based on this, the paper adopts object-oriented image classification technology to interpret and extract UTC data from satellite images, and it uses house price to explore the justice of the spatial distribution of UTC in the main urban area of Guangzhou from the perspective of environmental justice by ANOVA, Pearson correlation coefficient, and bivariate local spatial autocorrelation. The results show that: (1) There is a significant positive correlation between UTC and house price in the main urban area of Guangzhou, and there are regional differences in the distribution of UTC, with the UTC in the “very high” house price group being significantly higher than that in other groups. (2) The spatial clustering of UTC and house price in the main urban area of Guangzhou is found to be “low-low” and “high-high”; thus, it can be concluded that the spatial distribution of UTC in the main urban area of Guangzhou is uneven. This is an environmental injustice, as the areas with low UTC are spatially clustered in old residential areas, while the areas with high UTC are spatially clustered in commercial housing estates with high house prices. The study shows that urban tree planting should not only focus on quantitative improvement but also on equitable spatial layouts to promote social equity and justice thereby improving the urban ecological environment and promoting healthy urban development.

## 1. Introduction

In the context of tightening global resource constraints, serious environmental pollution, and ecosystem degradation, healthy development of the urban ecological environment has become an important issue worldwide. In this regard, China is vigorously promoting the construction of ecological civilization and striving to build a beautiful home where people and nature coexist in harmony. In recent years, China’s urban habitat has improved significantly, with the green space rate in built-up areas reaching 38.7% in 2021. Urban green spaces provide ecosystem services for cities, alleviate urban heat island effects, improve air and water environment quality, and provide leisure activities and other functions [1]. Urban green spaces play a key role in ensuring urban ecological security, improving residents’ living quality, and maintaining health-conscious urban development [2]. As an indispensable and important part of urban green spaces, urban trees play key ecological functions, such as carbon absorption and storage, regulation of microclimate, elimination of atmospheric pollution, and noise reduction [3].

In western countries, urban tree canopy (UTC) has become an important indicator for urban tree resource assessment and is widely used in urban tree research and planning. The concept of UTC was born in the United States. UTC is the layer of leaves, branches, and stems of trees that cover the ground when viewed from above [4]. Many studies have shown that higher UTC not only provides ecological benefits such as reducing stormwater runoff [5], lowering ground temperature [6], and mitigating urban heat island effects [7], but also helps prevent and reduce the risk of various diseases, such as preventing Alzheimer’s disease [8], preventing heart disease [9], and reducing tuberculosis mortality [10]. In addition, increasing UTC contributes to reduced crime rates and a greater sense of safety in densely populated and poor areas [11]. For example, Austin Troy et al. found that a 10% increase in UTC was associated with an approximate 12% reduction in crime rates [12]. Although many cities around the world are working hard to improve the quality of life by increasing UTC, the spatial distribution of UTC is not evenly distributed across different areas of the same city. A study of the city of Atlanta, Georgia, USA showed that areas with lower levels of UTC had a higher percentage of African-American residents and relatively lower average household income than areas with high levels of UTC [13]. Kirsten Schwarz et al. showed that the distribution of UTC was significantly and positively correlated with residential income [14]. Additionally, Dexter H. Locke et al. found that over time, the Los Angeles, California, USA coast tended to have more increase and less loss of UTC in high-income areas [15]. For this reason, many scholars have explored the equitability of the spatial distribution of UTC from the perspective of environmental justice in recent years. Alec Foster et al. proposed that there was environmental injustice in the change of UTC in Philadelphia, Pennsylvania, USA from 2008 to 2018 [16], and Christopher S. Greene et al. studied Toronto, CA, and found significant injustice in the distribution of UTC [17]. Carl Kolosna and Danielle Spurlock argued that urban planning authorities are not considering or attempting to reduce the uneven spatial distribution of UTC [18]. At present, there are few applications of UTC in China, and the research in related fields started relatively late. Relevant research results have been published since 2011 [19,20,21]. The research contents mainly include the measurement of UTC [22], spatial characteristics of UTC [23], and dynamic changes of UTC [24], etc. The research sites are mainly Beijing, Shanghai, Guangzhou, and other megacities.

In recent years, the topic of environmental justice has been heavily covered by environmental ecology, environmental health, human geography, urban planning, and other disciplines. The research on environmental justice mainly includes three dimensions: distribution justice, recognition justice, and procedure justice [25]. From the perspective of distribution justice, it is important to ensure that every resident has equal access to various public facilities, and the equitable distribution of urban green spaces is also an important manifestation of environmental justice. However, related studies in China found that, in Tianjin, inequities existed among residents of different socioeconomic statuses in enjoying the health benefits of urban green spaces [26], and You Heyuan found that the number of public green spaces declined with district disadvantage, degree of income, occupation, and housing in Shenzhen [27]. In addition, the spatial distribution of urban green spaces in the central urban area of Hefei is also uneven [28]. Therefore, it can be seen that there is a problem of unequal distribution of urban green spaces in some cities in China.

Currently, some scholars are attempting to associate urban green space, or UTC, with socioeconomic status, since it can help identify the localities lacking urban green space or UTC supply for land use policymakers [29]. Nevertheless, socioeconomic data are often readily available at the city or regional level, whereas “micro” spatial scales, such as the community-, block-, or building-level, are more difficult to obtain, especially in developing countries and regions (e.g., mainland China) [30]. At present, most studies in China obtain house prices as socioeconomic data from real estate websites [31,32]. The real estate reforms of the past 20 years and the rise in house prices have led to the wealthy living in luxury commercial housing estates and the vulnerable group being left to live in old residential areas or resettlement housing estates. Thus, when paired with UTC data, the house price can fully reflect the socioeconomic status of city residents and the distributional equity of urban green spaces or UTC.

There are few studies related to the justice of the spatial distribution of UTC in China, and less attention has been paid to the relationship between the distribution of UTC within Chinese cities and the socioeconomic status of residents. Compared with urban green spaces, UTC can quantify the trees on an impermeable surface, facilitating a more comprehensive assessment of the differences in the spatial distribution of green resources in different areas of the city. Based on this, the paper takes the main urban area of Guangzhou as the research area, adopts house price data as the evaluation index of socioeconomic status, and explores whether there is a link between UTC and house price through a spatial and non-spatial approach from the perspective of environmental justice. Specifically, the objectives of this study are to evaluate the following three aspects: (1) UTC in the main urban area of Guangzhou. (2) Whether there are significant regional differences between house price and UTC distribution. (3) Whether a spatial correlation exists between house price and UTC.

## 2. Research Area

Guangzhou, located in southern China, is the capital of Guangdong Province, which was founded in 214 BC and has a long history of about 2220 years. In recent years, Guangzhou has attached importance to the construction of urban green space and the improvement of the living environment and has become one of China’s happiest cities. In 2021, Guangzhou had a forest coverage rate of 41.6%, a built-up area green coverage rate of 45.52%, and the first green spaces per capita in the country. In the future, Guangzhou will strive to narrow the gap between urban and rural development and residents’ living standards and build a low-carbon and environmentally friendly international metropolis. 

However, some scholars found that in more than half of the neighborhoods in Guangzhou, the older adults, less-educated population, immigrants, and households with a living area below 50 m^2^ were suffering from urban green space inequity [33]. Moreover, the distribution, accessibility, and frequency of usage of urban green spaces, as well as individual demographic and socioeconomic factors, are strongly associated with health issues in Guangzhou [34]. Given the inequities in urban green space distribution observed in Guangzhou by previous research and the demonstrated importance of urban green space access to human health and wellbeing, Guangzhou is an ideal case study city to examine UTC from an environmental justice lens. Following the regional division of Guangzhou in relevant studies [35], and considering the influence of some mountainous, woodland, and rural areas, the paper defines the scope of the main urban area of Guangzhou as “old urban area + core area” and takes it as the research area (Figure 1). The research area covers an area of 86.19 km^2^, including part of Yuexiu District, Liwan District, Haizhu District, and Tianhe District.

At present, UTC in China is not available from relevant departments, and UTC data are mainly obtained through high-resolution satellite image interpretation. Since the spatial resolution of most high-resolution satellite images is between 0.5–1 m, it is difficult to extract shrubs and trees separately at this resolution. In addition, the forest law of China enforces that certain regions, known as “national special provisions shrub forests”, grow only shrub vegetation due to their unique climate and land conditions. These regions were included in the calculation of the legal forest coverage rate of China. Because shrubs in this region have the same status as trees, both trees and shrubs are included in the UTC. According to the specific conditions of the research area, land use types are divided into six categories: impermeable surface, water body, trees and shrubs, grass, and bare land, as shown in Table 1.

Since there were only 48 street units in the research area, the sample size was too small if streets were used as units, and the research results lacked statistical significance. Based on this, the paper manually arranges the sample land in the research area according to a grid and divides the sample land into 500 m × 500 m cells in the research area according to the 5-min pedestrian-scale neighborhood (about 23 ha) from the Design Standards for Urban Residential Area Division [36]. Each sample land cell can be regarded as a 5-min pedestrian-scale neighborhood.

## 3. Research Methods and Data Sources

Based on the perspective of environmental justice, the paper takes the main urban area of Guangzhou as the research area and uses the data of UTC and house prices to examine the distributional equity of UTC. In terms of data sources, the paper uses Python to extract the house price transaction data and its spatial coordinates in Guangzhou from January 2021 to December 2021 from the Lianjia real estate transaction website (https://gz.lianjia.com/, accessed on 28 September 2022). The panchromatic satellite image from Google with a spatial resolution of 0.5 m in 2021 is selected as the UTC data, and we use object-oriented image classification technology to interpret the image and identify the UTC. Current studies typically apply simple statistical methods, such as Pearson correlation coefficient, Spearman’s correlation, ordinary least squares regression, etc., but these methods lack consideration of the spatial impact of each factor [29]. Hence, this paper employs analysis of variance (ANOVA), Pearson correlation coefficient, and bivariate local spatial autocorrelation at the statistical and spatial levels to examine whether there is an environmental injustice between the UTC and the house price. The specific research process is shown in Figure 2.

In this paper, eCognition is used to interpret and extract UTC data from satellite images. ECognition is an image analysis software developed by Definiens Imaging in Germany in 2000 with powerful object-oriented image classification technology. It can process and analyze medium-high resolution satellite images, remote sensing images, spectral images, and other image data, and it is widely used in urban planning, forest protection and management, land use, natural disaster monitoring, and other fields [37]. Object-oriented image classification technology includes two modules: the image segmentation module and the image object classification module. In this paper, multi-scale segmentation and nearest-neighbor classification are used for image processing. The multi-scale segmentation parameters include the compactness parameter, shape parameter, and segmentation scale. By referring to relevant literature [38,39] and repeated testing, the optimal segmentation parameters of each land use type are determined (Table 2).

To more accurately examine the correlation between UTC and house price, the paper excludes grids without house price data to reduce interference with the analysis results. The paper calculates the average house price for each sample location separately and classifies the house price into four sample groups: low, medium, high, and very high (Table 3). At the statistical level, ANOVA and Pearson’s coefficient analysis are performed using SPSS. In studies related to UTC and socioeconomic status, ANOVA and Pearson coefficient analysis are commonly used as statistical methods [17,40]. The ANOVA measures the differences between groups of data as the dispersion of the mean UTC for each group of house price samples, and thus examines whether UTC is significantly different among the four sample groups with different classes of house price. If there is a large difference in the mean value of UTC across the different house price sample groups, it indicates that the UTC distribution is uneven. The Pearson correlation coefficient is measured by the average house price of each grid sample site and its corresponding UTC; thus, it tests the linear correlation between the UTC and the average house price. The value of the Pearson correlation coefficient is between −1 and 1. The larger the absolute value is, the stronger the correlation.

At the spatial level, Geoda is used to conduct the bivariate local spatial autocorrelation analysis. Spatial autocorrelation refers to the statistical correlation between the values of an attribute of a geographical feature distributed across different spatial locations, and usually, the closer the distance is between or among features, the stronger the correlation is among the attribute values. The local spatial autocorrelation analysis produces a Moran’s I metric, whose value is between −1 and 1. Larger absolute values indicate stronger spatial autocorrelation [41]. Spatial autocorrelation is divided into global and local spatial autocorrelation. Global spatial autocorrelation refers to the spatial features in the whole region, while local spatial autocorrelation refers to the correlation of each spatial location in the region with its respective surrounding neighboring locations [42]. The paper uses bivariate local spatial autocorrelation to explore the local correlation between house price and UTC, with five spatial distribution patterns of high-high aggregation, low-low aggregation, low-high aggregation, high-low aggregation, and not significant. Among them, high-high aggregation means that the house price and UTC of this region and its neighboring areas are both high, while low-low aggregation means that the house price and UTC of this region and its neighboring areas are both low.

## 4. Research Results

### 4.1. UTC

UTC reflects the current tree resource retention and is an important indicator of the quality of the urban ecological environment. The distribution of land use types in the main urban area of Guangzhou is shown in Figure 3, in which the total area covered by trees and shrubs is 21.14 km^2^, and the UTC is 24.52% (Table 4). Parks in the main urban area of Guangzhou have an absolute advantage in UTC, including Yuexiu Park, Liwan Lake Park, Guangzhou Martyr Memorial Park, Tianhe Park, and Yellow Flower Mound Park. They are the main contributors to high-density green space in Guangzhou. In addition, street trees are an integral part of the UTC contribution in the main city of Guangzhou. UTC in residential areas is mainly concentrated in commercial housing estates, but UTC in old residential areas and urban villages is low. Moreover, Jinan University, Sun Yat-sen University, and the Guangzhou Zoo also provide plenty of UTC. In terms of other land use types, impermeable surface accounts for 63.11%, which is related to the high development utilization rate of the main urban area of Guangzhou. The coverage of water bodies is 7.51%, mainly in Pearl River water. The coverage of bare land is 2.27%, which includes bare soil surface during the construction period and a small amount of unused land. Grass coverage is 1.92%, mainly in the vicinity of Ersha Island and Guangzhou Tower.

### 4.2. The Relationship between House Price and UTC

#### 4.2.1. Statistical Analysis

ANOVA provides a way to examine whether there is a significant difference in the mean UTC for each house-price-class sample. As shown in Table 5, the mean difference of UTC for each house-price-class sample is significant, and the mean difference of some groups has a high significance level. The results of the study show that the mean difference of UTC in the main urban area of Guangzhou is positively correlated with the house price class. In addition, as the house price class increases, the mean difference of UTC increases for most of the sample. Among them, the mean difference of UTC in the “very high” group is significantly higher than that in the other groups, with significance levels of 0.001, and compared to the “low” group, the mean difference in UTC is as high as 12.85%. It can be seen that the UTC in the “very high” group of the main urban area of Guangzhou is significantly higher than that in other areas, and there is a clear polarization phenomenon such that UTC levels are highest in high housing price classes and lowest in low housing price classes. However, the mean difference between the “medium” and “high” groups is low, with a mean difference of 0.50%. This result may be related to the way the paper classifies house price classes and the effect of other factors on house prices in the “high” group.

The Pearson correlation coefficient can be used to test the linear correlation between average house price and UTC. As shown in Table 6, the correlation between average house price and UTC is significant, with a Pearson correlation coefficient of 0.375. That is, there is a significant positive correlation between the average house price and UTC in the main urban area of Guangzhou, and as UTC increases, the average house price also increases significantly.

#### 4.2.2. Spatial Analysis

Bivariate local spatial autocorrelation is used to test the degree of aggregation between house price and UTC within a local area, with five spatial distribution patterns: high-high aggregation, low-low aggregation, low-high aggregation, high-low aggregation, and not significant. As shown in Figure 4, Moran’s I of UTC and average house price is 0.245, which means that there is a positive spatial aggregation of UTC and average house price in the main urban area of Guangzhou.

The spatial distribution of low-low aggregation is mainly located in cluster I and cluster II, both of which belong to the old urban area of Guangzhou. Cluster I is the historical and cultural district of Liwan District and Haizhu District of Guangzhou, including Zhongshang 8th Road, Changshou Road, Hualin Temple, Chen Clan Academy, Tongfuxi, and other areas. Cluster II is an old residential area in Shaheding, Yuexiu District. The spatial distribution of high-high aggregation is mainly located in cluster III and cluster IV, which are mostly commercial housing estates. Cluster III is the main central axis of Guangzhou and part of the area along the Pearl River, which is one of the very prosperous areas of Guangzhou, including Ersha Island, Liede, Pearl River New City, Pearl River Park, and other areas. In addition, cluster IV is located at Guangzhou East Railway Station and Linhexi, both of which are important transportation hubs in Guangzhou. It is worth noting that the high-low aggregation appears in cluster V, or Jiangnanxi, which is also the old urban area. However, compared to the old urban area described above, Jiangnanxi is close to Xiaogang Park and has a high street tree cover. Moreover, cluster VI shows a low-high aggregation, and it has higher house prices but lower UTC. This may be due to the high accessibility of public transportation stations in this area and the concentration of amenities such as schools, hospitals, and shopping malls, leading to higher house prices.

## 5. Discussion

Compared with traditional indicators such as urban green rate, urban green coverage rate, and the per capita green area of the park in China, UTC can not only objectively and accurately reflect the current number of urban trees but also can be used to evaluate and monitor urban tree construction and propose corresponding planning objectives. Many western cities set targets for UTC when preparing urban planning proposals. Colombia, South Carolina, USA have set a goal of 40% UTC by 2032 and has set corresponding planting targets for governments, nonprofits, for-profit organizations, and individuals. Vancouver, CA has a goal of 28% UTC by 2030, and Hartford, Connecticut, USA has also set a goal of 35% UTC within 50 years.

In this paper, ANOVA, Pearson correlation coefficient, and bivariate local spatial autocorrelation are used to examine whether there is a relationship between UTC and house price in the main urban area of Guangzhou. The research shows that the distribution of UTC in different house price areas in the main urban area of Guangzhou is uneven. The phenomenon of “low-low” and “high-high” spatial aggregations are present, indicating that there is an environmental injustice problem, with low-house-price areas experiencing significantly lower UTC than high-house-price areas. The main reason for this phenomenon is that the areas with high house prices are generally located in commercial housing estates, which have well-designed greening landscapes and comfortable green living environments. Nowadays, residents are looking for urban trees and urban green spaces, which has led to very high house prices in these places. It has even led to environmental gentrification in some high UTC areas, resulting in the displacement of vulnerable groups. Furthermore, due to the dense buildings, limited tree planting space, sparse public parcels, and limited access to power or capital for planting and maintaining trees in low-house price areas within Guangzhou such as old residential areas, urban villages, and resettlement housing estates, these areas generally have low UTC and poor tree health. As can be seen, the distribution of UTC in Guangzhou is greatly affected by house prices and even socioeconomic status.

Studies have found similar patterns in some cities; in Milwaukee, Minnesota, USA, low-income and minority populations mostly live in areas with low UTC [40]. The distribution of UTC in Cali, CO is related to the socioeconomic status of residents [43], and the UTC of commercial housing in Beijing, CHN is about 10% higher than that of temporary dwellings [24]. Additionally, Huang Shuaishuai found that the tree health index of commercial housing areas in Beijing, CHN is higher than that of security housing [44]. Despite these findings, many cities are currently working to improve the urban green rate and UTC to regulate the urban climate, improve the ecological environment, promote the quality of life, and enhance the happiness of residents. However, the uneven distribution of UTC leads to an uneven quality of life among city residents, and it is difficult for low-income people and marginalized populations to equitably obtain the benefits provided by urban trees. Demand from residents for parks and urban green spaces has increased since the COVID-19 pandemic outbreak began and highlights the important role and benefits provided by parks, especially urban and community parks [45]. Moreover, Christopher S. Greene et al. also proposed that we should not only consider the spatial distribution of the benefits provided by the urban ecological environment in the short term (regional equity), but also consider the sustainability of these benefits, and we must ensure that these benefits are shared by our children and grandchildren (inter-generational equity), especially the climate benefits they bring [17]. Only by fairly obtaining various benefits provided by the urban ecological environment and ensuring the just and reasonable distribution of urban green spaces in each area of the city, can we ensure that every resident has a comfortable, healthy, and safe environment.

Although the study of UTC can elucidate the differences in the distribution of urban trees among different regions, it cannot substantively solve the problem of environmental justice. Lorien Nesbitt et al. argued that the focus of current environmental justice research is distribution justice, while procedure justice remains unresolved [46]. Procedure justice refers to fairness and justice in the execution of procedures or judgments. In this process, as vulnerable groups often lack the right to speak, environmental improvement will only perpetuate or produce new spatial inequalities and injustices [47]. Therefore, in future urban tree-planting planning, relevant departments should follow the principle of fairness and justice and give priority to areas where low-income people and marginalized populations are located. To be specific, we believe that Guangzhou should establish minimum guarantee standards for UTC and incorporate them into the code for the planning and design of urban residential areas. Technically, it is necessary to develop an effective supervisory management tool that can spatially locate areas that lack trees or require tree maintenance. Such a tool can facilitate the spatial prioritization of tree planting planning. Furthermore, policymakers and planners could also give these populations a greater say by involving local residents in the planning, decision-making, and implementation processes. To solve the problem of procedure injustice in the process of environmental change, the key lies in breaking the unequal relationship between the political and economic environment and power, rather than focusing on the environment itself.

Due to data limitations, the paper has the following deficiencies: (1) House price is used as an evaluation index of socioeconomic status, but areas with low house prices may partially cover high-income or high-social-class groups, which affects the accuracy of the results to a certain extent. (2) Dividing the sample land into 500 m × 500 m cells in the research area may lead to the segmentation of part of the residential neighborhood. If a street unit or neighborhood unit is used for division, the house price and UTC of each area can be more accurately matched at these finer scales. (3) The paper takes 2021 as the time point for the study. Because tree growth and cultivation area dynamic processes, the inclusion of additional time periods can provide a fuller understanding of the changes in UTC.

## 6. Conclusions

The paper investigates the relationship between UTC and house prices in the main urban area of Guangzhou through statistical and spatial analysis from the perspective of environmental justice. The study finds that: (1) The UTC in the main urban area of Guangzhou is 24.52%. Compared with other cities in China (UTC in the urban region of Xiamen: 25.71% [22]; UTC in the urban residential areas of Beijing: 29.67% [23]; UTC in the built-up area of Tengchong: 38.90% [48]), the UTC in the main urban area of Guangzhou is in the middle to lower range, and it needs to be further improved in the future. (2) There is a significant positive correlation between UTC and house price in the main urban area of Guangzhou, and there are regional differences in the distribution of UTC, with the “very high” group having a significantly higher UTC than other areas. (3) There is a local spatial aggregation of UTC and house prices in the main urban area of Guangzhou, mainly between “low-low” aggregation and “high-high” aggregation. Statistically, the distribution of UTC is significantly polarized, and the UTC of the “very high” group is significantly higher than that of the “low” group. Moreover, the areas with low UTC are spatially clustered in old residential areas, while the areas with high UTC are spatially clustered in commercial housing estates with high house prices. In summary, it can be concluded that the spatial distribution of UTC is uneven in the main urban area of Guangzhou. The UTC is high in areas with high house prices and low in areas with low house prices, which indicates the existence of environmental injustice.

The study of UTC from the perspective of environmental justice is helpful to reveal the relationship between the urban ecological environment and urban social space, and it can improve the understanding of equity and justice to promote healthy urban development. The article suggests three potential future research directions to further advance the study of environmental justice and UTC. (1) By combining land use type, per capita income, education level, health index, and other data, future work can conduct in-depth analyses of the spatial structure of urban trees to expand the research on distribution justice. (2) From the perspective of procedure justice, we should increase the investigation of local economic conditions, social background, and historical background of neighborhoods and residents. (3) Through qualitative research, we should explore the fairness and justice of urban tree planting from the micro perspective, focusing on finer-scale assessments. The benefits of urban green space access are well documented; thus, we should strive to optimize the spatial distribution of urban green spaces to promote social equity and justice and make “clear waters and green mountains” benefit all people fairly.

## Figures and Tables

**Figure 1 ijerph-20-04050-f001:**
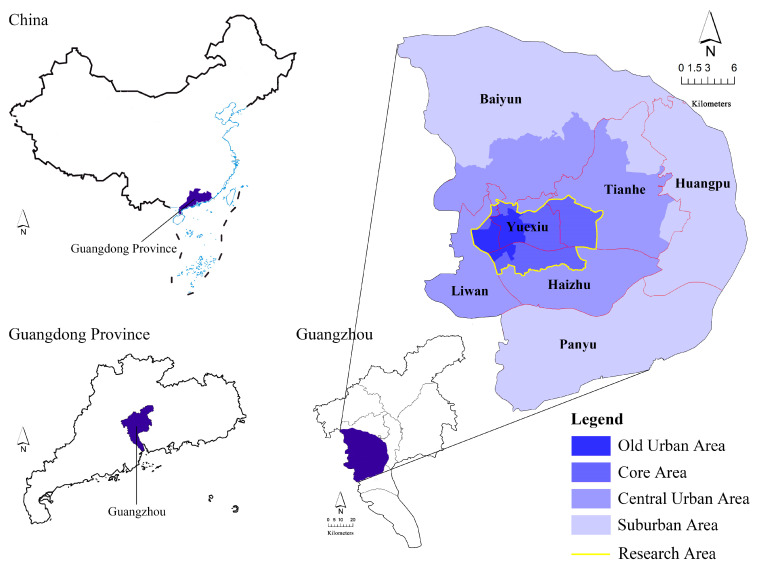
Map of the research area.

**Figure 2 ijerph-20-04050-f002:**
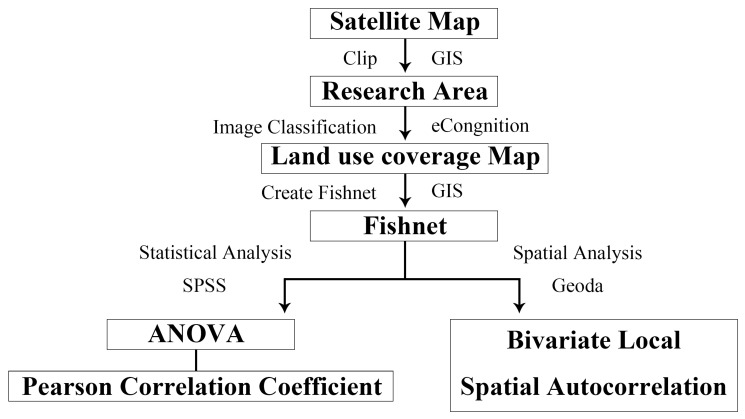
Research process.

**Figure 3 ijerph-20-04050-f003:**
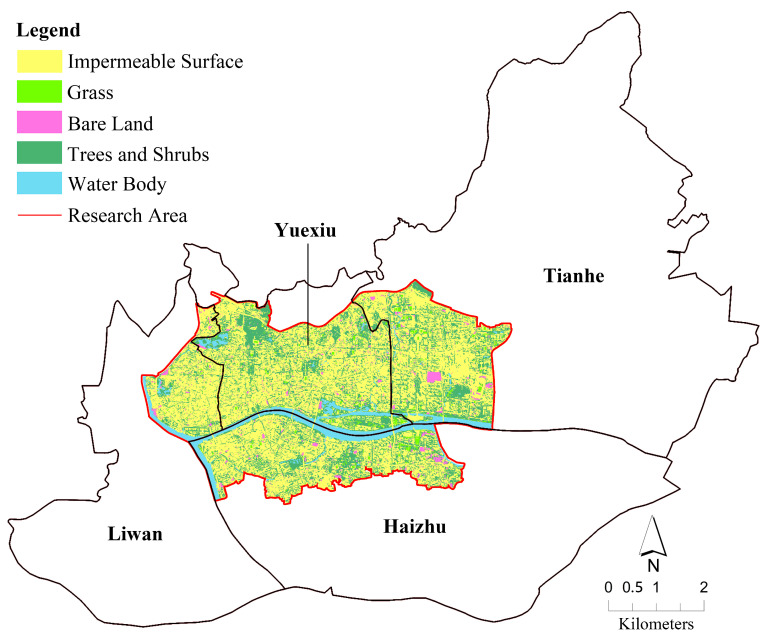
Land use coverage map of the main urban area of Guangzhou.

**Figure 4 ijerph-20-04050-f004:**
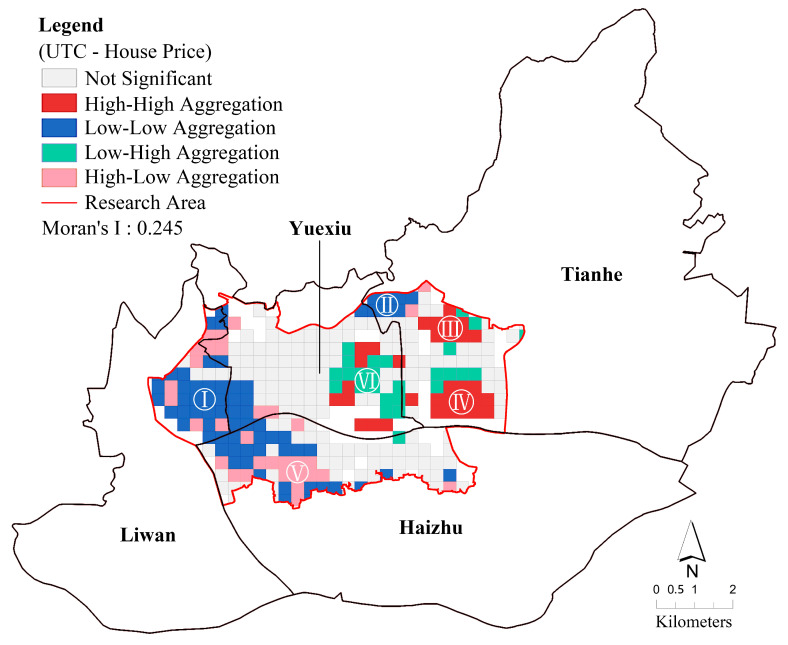
Bivariate local spatial autocorrelation analysis of UTC and house price.

**Table 1 ijerph-20-04050-t001:** Land use classifications.

Types	Description
Impermeable Surface	Artificial surfaces where water cannot penetrate directly into the soil, including roads, parking lots, building roofs, squares, etc.
Water Body	Including rivers, lakes, canals, ponds, open-air swimming pools, etc.
Trees and Shrubs	Erect woody plants, including clusters as well as isolated canopies of trees and shrubs.
Grass	Land with a surface layer of soil, growing herbaceous plants, and more than 10% vegetation cover.
Bare Land	Land with a surface layer of soil and no vegetation cover or less than 10% vegetation cover.

**Table 2 ijerph-20-04050-t002:** Optimal segmentation parameters.

Types	Shape Parameter	Compactness Parameter	Segmentation Scale
Impermeable Surface	0.4	0.8	150
Water Body	0.2	0.5	200
Trees and Shrubs	0.2	0.6	50
Grass	0.4	0.6	80
Bare Land	0.4	0.8	150

**Table 3 ijerph-20-04050-t003:** Sample house price divisions.

	House Price Group	Average House Price (RMB)	Sample Size
1	Low	0–35,000	71
2	Medium	35,000–50,000	112
3	High	500,000–65,000	80
4	Very high	>65,000	48

**Table 4 ijerph-20-04050-t004:** Statistical table of land use coverage in the main urban area of Guangzhou.

Land Use Types	Area (km^2^)	Proportion (%)
Impermeable Surface	54.39	63.11
Trees and Shrubs	21.14	24.52
Water Body	6.47	7.51
Bare Land	2.27	2.64
Grass	1.92	2.22
Total	86.19	100.00

**Table 5 ijerph-20-04050-t005:** Analysis of variance table.

House Price Group (I)	House Price Group (J)	Mean Difference (I–J)
		UTC (%)
1	2	−2.70
	3	−3.25 *
	4	−12.85 **
2	1	2.70
	3	−0.50
	4	−10.15 **
3	1	3.25 *
	2	0.50
	4	−9.61 **
4	1	12.85 **
	2	10.15 **
	3	9.61 **

* *p* < 0.05, ** *p* < 0.001.

**Table 6 ijerph-20-04050-t006:** Pearson correlation coefficient table.

	Average House Price	UTC
Average House Price	1	0.375 *
UTC	0.375 *	1

* *p* < 0.001.

## Data Availability

Not applicable.

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
