# Peer review of "Urban Tree Canopy and Environmental Justice: Examining the Distributional Equity of Urban Tree Canopy in Guangzhou, China"

_ijerph, 2023, doi:10.3390/ijerph20054050_

Round 1

Reviewer 1 Report

The title of this paper is very attractive to me. But after I read the text of the paper, I felt that I didn't agree with it.

1. What factors contribute to the fairness of tree canopy distribution? The paper can be summarized into the types in Table 1. I think these are not enough. Many are from government factors.

2. I also have doubts about the research method of the paper. Why use ANOVA? Is multifactor analysis appropriate? Or other methods.

3. In the third part, what is the relationship between the real estate transaction data and this topic? I'm confused! This dimension is also not mentioned in the previous review, so it is suggested to comb the argument carefully.

4. It is a good way to use software analysis, but it should match with the research topic. I think there is a problem between the research method and the theme of this paper.

5. A lot of English grammar needs to be revised.

I think this paper needs to be adjusted from the methodological level. A lot of modifications are required.

Reviewer 2 Report

The paper is the study of urban tree canopy in Guangzhou, China.

The manuscript is rather correctly structured and referenced. However, there is no need to repeat the aim of the study twice (lines 97-100 in the Introduction and lines 140-145 in the Research methods).

Also in the Results section the table 5 is not self-explanatory – what are J and I ?

The authors observed that house prices are higher in the more green locations of the examined city. But they do not provide readers with the crucial pieces of information:

How old is the city of Guangzhou? How was it developing so far and what are the plans for its future development?

I would advise to re-write ‘Discussion’ in order to interpret the results on the basis of those data. It is still unclear whether the prices in some zones of Guangzhou are so high because its inhabitants really seek urban green spaces or because properties are bigger or constructed from the better quality materials etc.

The authors also mention (lines 340-345) that ‘in future urban tree planting planning, relevant departments should follow the principle of fairness and justice’. It would be interesting and useful if the authors would try to answer whether it is technically possible to improve UTC coverage in the areas of present low UTC in Guangzhou or are they to remain inhospitable forever.

Round 2

Reviewer 1 Report

Thank you for your revision. I have no new suggestions.